# Downstream Signalling from Molecular Hydrogen

**DOI:** 10.3390/plants10020367

**Published:** 2021-02-14

**Authors:** John T. Hancock, Grace Russell

**Affiliations:** Department of Applied Sciences, University of the West of England, Bristol BS16 1QY, UK; Grace2.Russell@live.uwe.ac.uk

**Keywords:** antioxidants, heme oxygenase, hydrogen gas, hydrogenase, hydroxyl radicals, molecular hydrogen, nitric oxide, reactive oxygen species

## Abstract

Molecular hydrogen (H_2_) is now considered part of the suite of small molecules that can control cellular activity. As such, H_2_ has been suggested to be used in the therapy of diseases in humans and in plant science to enhance the growth and productivity of plants. Treatments of plants may involve the creation of hydrogen-rich water (HRW), which can then be applied to the foliage or roots systems of the plants. However, the molecular action of H_2_ remains elusive. It has been suggested that the presence of H_2_ may act as an antioxidant or on the antioxidant capacity of cells, perhaps through the scavenging of hydroxyl radicals. H_2_ may act through influencing heme oxygenase activity or through the interaction with reactive nitrogen species. However, controversy exists around all the mechanisms suggested. Here, the downstream mechanisms in which H_2_ may be involved are critically reviewed, with a particular emphasis on the H_2_ mitigation of stress responses. Hopefully, this review will provide insight that may inform future research in this area.

## 1. Introduction

Molecular hydrogen (H_2_) is now recognized to have biochemical effects in both animals [1,2] and plants [3,4]. Although it is a relatively inert gas, H_2_ appears to have profound effects on cell activity, which can be harnessed to help plant growth, survival, and productivity [5,6,7,8].

Plants, particularly as they are sessile, have to endure and survive a wide range of stress challenges, both biotic and abiotic. These stresses include attack by pathogens [9] and insects [10], as well as heavy metals [11], extreme temperature [12], salt [13], and ultraviolet B light [14]. It has become apparent over many years of study that there are common molecular responses to such stresses, and these mechanisms often involve reactive oxygen species (ROS) [15] and reactive nitrogen species (RNS) [16]. These compounds include ROS such as superoxide anions (O_2_·^−^) and hydrogen peroxide (H_2_O_2_), the latter of which is a major focus of ROS signalling [17]. Importantly, ROS also include the hydroxyl radical (·OH). The most prominent RNS is nitric oxide (NO), which is known to be involved in plant cell signalling processes [18]. However, other RNS include peroxynitrite and nitrosoglutathione, both of which can act as signalling molecules [19,20]. It is also apparent that crosstalk occurs between ROS and RNS [21] as well as with other reactive signalling molecules such as hydrogen sulphide (H_2_S) [22,23].

H_2_ fits into this suite of reactive signalling molecules and was shown to increase the fitness of plants [24]. Suitable examples of recent papers on H_2_ effects on plants include mitigation of salinity effects in barley [25] and Arabidopsis [26], and increased tolerance to cadmium in alfalfa [27]. However, exactly how H_2_ interacts and has an effect is unclear. The metabolism of H_2_ in plants is not a novel idea [28] and some plants are known to be significant generators of H_2_, such as Chlamydomonas [29,30], whilst higher plants have been shown to produce H_2_ too. Plant H_2_ generation has been known for a long time [28,31], with more recent examples being reported using rice seedlings [32] and tomato plants [33]. The role of hydrogenase enzymes and the generation of H_2_ by plants was recently reviewed [7].

Molecular hydrogen, being a gas, is hard to use either in laboratory or environmental settings. It is extremely flammable [34], relatively insoluble [35,36], and will readily move to the gas phase. Despite this, treatment with H_2_ is often facilitated by the production of hydrogen-rich water (HRW), which can then be applied to the soil or directly onto the foliage. If using hydroponics, the HRW can be added directly to the feed solution. Several examples of the use of HRW are included throughout this review (for example, [5,8,37]). The use of HRW is effective and easy and is commonly used to treat plants, but treatment with H_2_ gas can also have cellular effects and is often used in animal studies, for example, with mice [38]. H_2_ gas has been used to alter plant growth by the gaseous treatment of the soil [39]. The treatment of biological materials with H_2_ was further discussed in previous papers [7,40].

Here, we provide a critical look at the correlation between the effect of H_2_ and the possible modes of action, with stress responses in plants being a focus. Issues that are addressed here include both the direct and indirect actions of H_2_ and what biological compounds H_2_ interacts within a cell, leading to the observed responses. Once this is established, a clearer view of downstream signal transduction initiated by H_2_ can be gained. It is hoped that this review will inform future research in this area of plant science. 

## 2. Downstream Effects

For any molecule to be used in cell signalling, it needs to be perceived by cells and to initiate a response. For many molecules, this involves a receptor protein, which may be on the cell surface [41] or in an intracellular compartment, such as the cytoplasm [42] or nucleus [43]. Some signalling molecules are perceived by proteins not classed as receptors, such as the effect of NO on soluble guanylyl cyclase (sGC). Here, NO reacts with the iron in the heme group of the enzyme, thereby activating it [44], although the involvement of such mechanisms has been questioned in plants [45]. Alternatively, the reactive nature of ROS and RNS allows them to oxidize [46] and nitrosate [47] thiol groups on proteins, propagating the signalling needed. It is hard to envisage how H_2_, being so small and relatively inert, can be perceived by cells. Some of the mechanisms reported and mooted are discussed below. 

### 2.1. Effects on Reactive Oxygen Species and Antioxidant Capacity

Stress responses in plants often involve ROS metabolism. There is often an increase in ROS accumulation, which, in some cases, can initiate programmed cell death (PCD) in plants [48]. ROS accumulate in the presence of heavy metals [49], such as cadmium [50], mercury, and copper [51]. ROS also accumulate in the presence of salt, extreme temperature, and pathogens [52]. Increases in the intracellular ROS under such stress conditions are often accompanied by an increase in antioxidant levels in cells, for example, in the presence of salt [53], heavy metals [54], and extreme temperature [55]. Therefore, the modulation of ROS metabolism is crucial for stress responses: increases in ROS lead to changes in cellular function, whilst antioxidants modulate and dampen that response. 

H_2_ has been shown to be able to help plant cells mitigate stress challenge. H_2_ can help reduce salt stress [56,57], and reduce stress due to aluminium [58,59], cadmium [60], and mercury [61]. H_2_ also can help mitigate against drought stress [62,63] and paraquat induced oxidative stress [64]. 

Xie et al. [57] suggested that H_2_ modulates plant cells’ antioxidant capacity through acting through zinc-finger transcription factor ZAT10/12. This would dampen the ROS accumulation and associated lipid peroxidation. They also suggested that H_2_ would act on the antiporters and proton pumps responsible for exclusion of Na^+^, particularly the protein salt overly sensitive1 (SOS1). Finally, it was suggested that both SOS1 and cytosolic ascorbate peroxidase1 (cAPX1) are molecular targets of H_2_-mediated signalling. Additionally, Xu et al. [59] also suggested that H_2_ may alter gene expression. In a study of aluminium stress, they found that H_2_ altered the ratio of gibberellin acid (GA) and abscisic acid (ABA), with the expression of genes for GA biosynthesis (*GA20ox1* and *GA20ox2*) and for ABA breakdown (*ABA8ox1* and *ABA8ox2*) being induced by H_2_. H_2_ also altered miRNA expression with downstream effects that increased superoxide dismutase (SOD) expression, increasing antioxidant levels in the cells. However, even though these findings all support the notion that H_2_ is protecting the cells, no direct interaction with H_2_ has been established. 

As can be seen from the discussion above, both stress responses and the effects of H_2_ can be linked to ROS metabolism and antioxidant levels in cells. Therefore, it is particularly pertinent that H_2_ has been posited to be an antioxidant [65]. Although this study discusses the effects in H_2_ in a clinical setting, the redox chemistry would be the same in plants cells. In an animal setting, a study showed that H_2_ is an antioxidant against the hydroxyl radical (·OH) but has no effects against other ROS [66]. This is most significant, as it is usually hydrogen peroxide (H_2_O_2_) that is deemed to be the primary inter- and intracellular signal [17,67]. Of importance, the specificity of H_2_ to scavenge ·OH has been disputed, as an in vitro study showed that H_2_ can scavenge H_2_O_2_. However, H_2_ could not scavenge superoxide anions [57]. In an experiment looking at the radiolysis of water, a negligible effect on the formation or consumption of H_2_O_2_ was seen when molecular hydrogen was added [68].

If, as suggested [66], the effects of H_2_ are mediated partly by ·OH scavenging, a series of questions could be asked: How influential are the levels of hydroxyl radicals in cells, and could H_2_ be acting through their modulation? Would this account for the effects seen? 

Hydroxyl radicals are known to have effects in plant cells. Richards et al. [69] described the hydroxyl radical as being a “potent regulator in plant cell biology”. They discussed the role of this molecule in numerous physiological mechanisms in plants, including germination, control of stomatal apertures, reproduction, and adaptation to stress challenge. ·OH has also been shown to be important for ion currents in roots [70,71]. In animal cells, ·OH was shown to be upstream of mitogen-activated protein kinases (MAPKs) and transcription factors (ERK2 and NF-κB) [72], and analogous mechanisms could exist in plants. Therefore, evidence exists of ·OH acting in a positive cell signalling role, which could potentially be the target of H_2_.

In biological systems, ROS are often the product of the sequential reduction of molecular oxygen, resulting ultimately in the 4-electron reduction to water (Equation (1)).
(1)$$O_{2}\;\overset{e^{-}}{\to }\;{O_{2}}^{-}\;\overset{e^{-}}{\to }\;H_{2}O_{2}\;\overset{e^{-}}{\to }\;2(·\mathrm{OH})\;\overset{e^{-}}{\to }\;2H_{2}O$$

The superoxide anion (O_2_·^–^) can be produced enzymatically, for example from the action of NADPH oxidases [73]. H_2_O_2_ can be produced by the subsequent dismutation of O_2_·^–^ by the enzyme family of superoxide dismutases (SOD) [74]. 

·OH can be then be subsequently produced, especially in the presence of metal ions [75,76]. This generation can be either from the Fenton reaction from H_2_O_2_ (Equation (2)):H_2_O_2_ + Fe^2^^+^ → ·OH + HO^–^ + Fe^3^^+^(2)

Or in the presence of transition metals through the Haber–Weiss reaction, using superoxide anions and H_2_O_2_ (Equation (3)):H_2_O_2_ + O_2_^–^ → ·OH + OH^–^ + O_2_(3)

If the production of ROS is initiated, for example, during a stress response as discussed above, the generation of ·OH is likely to proceed. Hydroxyl radicals can be detected in plant cells [77,78], and have been found to have multiple effects.

The application of H_2_ has mitigating influences during stress, and therefore if the effects of H_2_ are mediated by the removal of ·OH, then it might be expected that ·OH radicals would need to be produced during these stress responses, assuming H_2_ is working in these cases as a ·OH scavenger. It is in fact the case that ·OH can be found in these circumstances. For example, hydroxyl radicals increase during metal ion challenge [79], a cellular challenge in which H_2_ has been shown to have a beneficial effect [58,59,60,61]. In a similar manner ·OH is produced during paraquat treatment of plants [80], another situation mitigated by H_2_ [64]. During chilling stress and drought stress, increases in free iron and H_2_O_2_ have been recorded, and this implicates hydroxyl radical generation in downstream cellular responses [81]. Once again, H_2_ has beneficial effects under drought conditions [62,63], as well as chilling stress [82]. ·OH and H_2_ also have similar actions in heat stress [83,84]. Therefore, it can be seen that there are many stress conditions which elicit accumulation of ·OH and are also relieved by the presence of H_2_, suggesting that the ·OH scavenging activity of H_2_ is potentially responsible for the changes in cellular activity seen. This of course does not consider any spatial-temporal differences in ·OH accumulation during different stresses, or plant species variations, but the correlation of ·OH action and H_2_ effects may be pointing to a possible mechanism. 

Certainly, to support the notion that ·OH removal by H_2_ could be biologically significant, a look at other ·OH scavengers may be useful. Such scavenging has been suggested to be useful for animal health [85], whilst in plants, mannitol has been suggested to be protective through this mechanism [81]. Sugars such as sucralose has been studied for its ·OH scavenging effects in Arabidopsis [86], whilst β-carboline alkaloids [87] and more novel compounds have been used in animal systems [88]. Such studies show that there is merit in modulating ·OH in cells, and therefore support the notion that such action by H_2_ may be significant.

On the other hand, and importantly, it has been suggested that the reaction of H_2_ with ·OH is too slow to be of physiological relevance [89], although the authors were discussing clinical settings. In this paper the rate constant for the reaction of H_2_ with ·OH producing H_2_O and H· is only 4.2 × 10^7^ M^−1^ s^−1^ (from [90,91]). The rate constant for other radical reactions was quoted as 10^9^ M^−1^ s^−1^. It was suggested [89] that the ·OH would react with other biomolecules before reacting with the H_2_, rendering the presence of H_2_ as being irrelevant. Others have doubted whether H_2_ has its effects through scavenging ·OH, although this is from a human health perspective [92]. Assuming this is correct, the correlation of ·OH production and H_2_ effects during stress responses would also be irrelevant, begging the question, if ·OH scavenging is not the mechanism, what is? 

It is possible that H_2_ has indirect effects on antioxidant levels. There are several reports of antioxidant levels in plant cells altering on H_2_ treatment. For example, this was reported in a study using black barley (*Hordeum distichum* L.) [93]. Antioxidant enzymes such as catalase and SOD were increased in maize [94] with similar effects in Chinese cabbage [95]. HRW was also found to maintain the intracellular redox status of plant cells through alterations the levels of reduced and oxidized glutathione (GSH and GSSG) [60]. However, the direct targets of H_2_ have not been identified in such studies. Therefore, it may be that H_2_ is having effects on the cells’ antioxidant capacity, which can be measured, but it may not be a direct effect on the ROS themselves. 

### 2.2. Impact on Reactive Nitrogen Species Metabolism

RNS, such as the nitric oxide radical (NO), have been known to have important effects in plant cells for over forty years [96], although there is still some controversy of their endogenous production and action [45]. NO, like ROS are well known to be involved in plant stress responses [97], many of which are ameliorated by H_2_ treatment, as discussed above. Therefore, the relationship between H_2_ presence and altered RNS metabolism is worth exploring. 

H_2_ has been shown to have effects in nitrogen fixation [98], although this is only one facet of this complex process. Nitrogen fixation relies on many factors including nutrient availability, the soil-plant interactions, and community facilitation as exemplified by the work carried out with the alpine shrub *Salix herbacea* [99,100,101]. H_2_ has also been shown to alter NO synthesis during auxin-mediated root growth [33]. Li et al. [102] reported that NO was involved in H_2_-induced root growth, whilst Zhu et al. [103] also link H_2_ and NO, reporting that H_2_ promoted NO accumulation through increases in the activities of possible synthesizing enzymes: NO synthase-like enzymes and nitrate reductase. Additionally, HRW increased NO accumulation in a study on stomatal closure [104]. On the other hand, HRW decreased NO accumulation in alfalfa [59].

It is likely that during a stress response NO and ROS are produced temporally and spatially together, and they can interact to produce downstream products. Superoxide anions and NO together can lead to the generation of the ·OH radical [105], and as discussed above this have been mooted as a potential mechanism of H_2_ action. However, superoxide anions and NO can react to produce peroxynitrite (ONOO^−^) [105], which can act as a signalling molecule in its own right [106,107], possibility through alterations of amino acids [108], with tyrosine nitration being a major covalent change seen [106] which could have important downstream effects [109].

It has been reported that H_2_ reacts with ONOO^−^, but not NO [66,110]. Therefore, it would be unlikely that H_2_ has direct effects in the NO signalling, *per se*. However, it was reported that H_2_ reacts with peroxynitrite, which would potentially alter NO-induced signalling pathways. Despite several papers discussing the scavenging of ONOO^−^ by H_2_ [58,60], it has been completely ruled out by others [89]. In this paper, as well as saying that the ·OH reaction is too slow, they report that H_2_: (1) does not alter the rate of conversion of ONOOH to NO_3_^−^ and H^+^; (2) does not alter the rates of ONOO^−^-mediated tyrosine nitration; (3) does not alter the oxidative stress responses mediated by either ONOO^−^ or ·OH. Therefore, even if effects on NO metabolism are seen, such as alterations in activities of synthesising enzymes, there appears to be no direct scavenging of RNS, or ·OH, by H_2_ which could account for the observed cellular effects. 

### 2.3. Stress, Heme Oxygenase and H_2_

An enzyme mechanism that has been found to be important for H_2_ effects in cells involves the heme oxygenase enzyme (HO-1). For example, this was shown to be involved in root development in cucumber on treatment with HRW [37]. Hydrogen-mediated tolerance to paraquat was also shown to involve heme oxygenase [64]. Similar data can be found in studies of animal systems, for example, in mice [111]. 

HO-1 has been shown to be involved in a range of abiotic stress responses in plants, including salt, heavy metals, UV light, and drought. Responses to stresses such as drought are complex, involving the result of many genes being expressed and the effects of gene polymorphisms, as seen with *Phaseolus vulgaris* L. [112,113,114,115], with wild types showing tolerance differences [116,117]. Resistance and tolerance to extreme temperatures are also important and involve complicated cellular responses [118,119,120,121]. Such responses are often associated with the accumulation of cellular ROS and RNS [120]. The catalytic action of HO-1 is the breakdown of heme. This is an oxygen-dependent reaction that uses NADPH as a cofactor and generates biliverdin, carbon monoxide (CO), and iron [121,122]. Interestingly, CO has been shown to be involved in signalling events in cells, and could mediate downstream effects of H_2_, whilst iron facilitates ·OH production, as discussed above.

However, no direct interaction between H_2_ and HO-1 seems to have been reported. Further, no reaction has been reported between H_2_ and CO in biological systems. Therefore, the connection between H_2_ treatment and alterations of HO-1 activity needs to be a focus for future research.

### 2.4. Paramagnetic Properties and Possible Cellular Effects

The above discussion throws doubt onto many biochemical and reactive aspects of H_2_ effects in cells. However, the physical properties of H_2_ may also be important. Hydrogen can exist with two nuclear spin states (ortho- and parahydrogen) [123,124]. It is the interconversion between these states that may be relevant here [125]. One of the interactions discussed was with NO, which could potentially alter NO signalling. There is also the possibility of interactions with transition metals [126]. This could have a potentially significant effect on cell signalling pathways, as many enzymes involved in signal transduction have metal prosthetic groups, including guanylyl cyclase (at least in animals), SOD, and many respiratory and photosynthetic components. Many of the aforementioned enzymes may be involved in ROS and RNS metabolism, which are important in plant responses to many stresses, with such conditions being mitigated by H_2_, as discussed above. It is conceivable that H_2_ may interact with the heme during the catalytic cycle of HO-1, accounting for the effects mediated by this enzyme.

This physical aspect of H_2_ action was mooted previously [127], although experimental evidence is lacking and future research may prove this avenue wrong. However, the idea of quantum biology is not confined to H_2_ effects, and the topic was recently reviewed [128]. It was suggested that biological processes may occur due to quantum mechanical effects. A more recent review on this topic was also published [129].

## 3. Discussion

H_2_ is known to be involved in the control of cellular functions in plant cells. For example, it was reported to be involved in both phytohormone signalling and stress responses [32]. On a pragmatic note, treatment with H_2_ in the form of HRW was suggested to be useful for delaying postharvest spoilage of fruit [5]. Therefore, it is known, like animal cells [1,130], that H_2_ has effects, and such actions may be harnessed for future manipulation of plant growth and crop enhancement [131].

Several mechanisms of H_2_ action have been suggested, as summarized in Figure 1.

One of the significant actions of H_2_ in biological systems was suggested to be its ·OH scavenging activity [66], as reported in animal systems [132]. A range of studies have shown that ·OH increases in cells under stressful conditions [79,80,81], whilst H_2_ has been shown to have effects on such stress responses [58,59,60,61]. It may be argued that removal of ·OH by H_2_, if it is involved in important ·OH signalling pathways, should be detrimental to cell function, although many studies have looked at scavenging ·OH as a beneficial approach to cell and organism health, both in plants and animals [81,85,86,87]. Hydroxyl radicals are extremely reactive, and react with kinetics that are diffusion-limited, with rate constants for a range of biomolecules being determined, including ATP and ADP [133]. ·OH radicals are known to react with proteins [134], which can lead to amino acid oxidation, crosslinking, and degradation of the polypeptide [135]. Lipids [136], carbohydrates [137], and DNA [138] are also ·OH targets. Therefore, the scavenging activity of H_2_ may prevent the harmful effects of ·OH, which may account for some of the observed effects. However, the biggest issue is the rate constant of the reaction between H_2_ and ·OH, which is deemed to be too slow for physiological relevance [89], suggesting that the other biomolecules may react first anyway, and therefore H_2_ would not influence the levels of oxidative stress. The same authors also ruled out reactions with peroxynitrite, as discussed above. Therefore, with H_2_ not able to scavenge other ROS [66] and the effects of H_2_ on both ·OH and ONOO^-^ being ruled out [89], it appears that the scavenging role of H_2_ may have limited effects in cells, at best.

Heme oxygenase is one enzyme that has been reported as mediating H_2_ effects [37,64]. Although being reported in several studies, as discussed above, there is little evidence of a direct interaction which could account for the data seen. However, not all the data are negative and seemingly point to dead ends. It was reported that H_2_ scavenged H_2_O_2_ [57], which, if confirmed and can be shown to have effects *in vivo*, would be very significant, as H_2_O_2_ is one of the major ROS signalling molecules [17,67]. However, in radiolysis experiments with H_2_O_2_, the addition of H_2_ only had a negligible effect [68], suggesting that more research in this area would be beneficial. Another positive effect that is worth exploring is the interaction of H_2_ with metals. It was suggested that the beneficial effects of H_2_ may be mediated by the reduction of Fe(III), oxidized as a result of oxidative stress. However, neither iron-sulphur clusters nor heme groups were reduced by the presence of H_2_ [89]. Even so, the effect of H_2_ on Fe(III) is an enticing suggestion, as transition metals are widely used in biological systems, making this is another area that merits further investigation. 

Finally, the paramagnetic properties of hydrogen may be relevant to its biological action, as previously mooted [127]. This may include interactions with NO or transition metals, but experimental data would be needed to support this notion. There are other papers with H_2_ in catalysis, but it is difficult to determine their relevance to biochemical reactions, as they are often conducted under non-physiological conditions, such as high pressure [139].

In conclusion, although the involvement of molecular hydrogen in plant function has been known for a long time [28], there is still considerable uncertainty surrounding the exact actions of H_2_ in cells. Its role as a direct antioxidant is doubted, although many cellular effects have been observed, including alterations in antioxidants, changes in enzyme activity, and modulation in gene expression. What is clear is that H_2_ may be useful for the mitigation of plant stress, so it has been proposed to have an exciting future [4,131].

## Figures and Tables

**Figure 1 plants-10-00367-f001:**
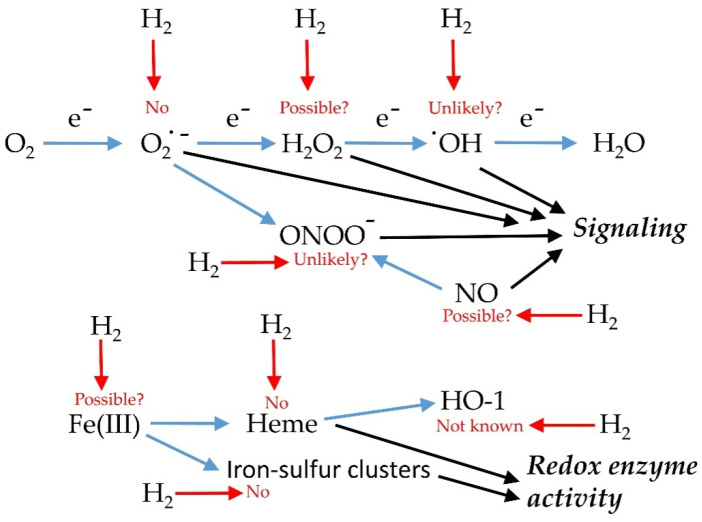
Possible mechanism of action of H_2_ in cells. The likelihood of there being effects on particular molecules is indicated (red arrows and text).

## Data Availability

There were no primary data generated in this study.

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
