# Peer review of "Downstream Signalling from Molecular Hydrogen"

_plants, 2021, doi:10.3390/plants10020367_

Round 1

Reviewer 1 Report

The manuscript by John T. Hancock entitled “Downstream Signaling from Molecular Hydrogen” has been reviewed. The authors declare that this review focused mainly on downstream mechanisms in which H2 may be involved, with a particular emphasis on H2 mitigation of stress responses. This review will give an interesting suggestion which may influence future research in Molecular hydrogen (H2) in plants. The correlation between the effect of H2 and the possible modes of action are elucidated in this review. This paper is more detailed about the content.

But, I will say that the relationship between H2 and ROS, and between H2 and RNS in plant cells, is very complicated (spatial-temporal change, in vitro and in vivo, and even different plant species/tissues and various stress conditions). So, it should be very cautious when writing. For example, some description in this review is too speculative (L149-L169, etc). Also, this MS lacks of latest papers of H2 mitigation of stress responses.

Other comments:

  1. L30 temperature changed to extreme temperature, light changed to Ultraviolet B light, and things are the same in other places, for example, L77 etc;
  2. L35 ROS also includes changed to ROS also include, and things are the same in other places, for example, L73 etc;
  3. L51water changed to solution; are should be deleted;
  4. L89 gene changed to molecular;
  5. L96 cells there changed to cells, there
  6. L105-L107 should be rewritten
  7. L148 L261 plants cells should be plant cells
  8. L152 heavy metal should be revised, since Al is not heavy metal (Ref51);
  9. L176 others too and this too?
  10. L205 Superoxide should be Superoxide anion;
  11. L231 to should be deleted
  12. L276-279 L290-L292 L304-L306 should be rewritten

Author Response

Response to Referees’ comments:

Referee 1:

Thank you for the feedback and comments on our manuscript. We are glad you liked it and we have addressed all the points raised, as outlined below. We hope that it is now acceptable for publication.

The manuscript by John T. Hancock entitled “Downstream Signaling from Molecular Hydrogen” has been reviewed. The authors declare that this review focused mainly on downstream mechanisms in which H2 may be involved, with a particular emphasis on H2 mitigation of stress responses. This review will give an interesting suggestion which may influence future research in Molecular hydrogen (H2) in plants. The correlation between the effect of H2 and the possible modes of action are elucidated in this review. This paper is more detailed about the content.

But, I will say that the relationship between H2 and ROS, and between H2 and RNS in plant cells, is very complicated (spatial-temporal change, in vitro and in vivo, and even different plant species/tissues and various stress conditions). So, it should be very cautious when writing. For example, some description in this review is too speculative (L149-L169, etc). Also, this MS lacks of latest papers of H2 mitigation of stress responses.

We fully appreciate this comment, but some speculation would be required for a review we feel. However, we have re-worded this section and added to it as suggested. We have also added very recent references for stress responses mitigated by H2 (line 42).

 Other comments:

  1. L30 temperature changed to extreme temperature, light changed to Ultraviolet B light, and things are the same in other places, for example, L77 etc;

Corrected as suggested, and the manuscript carefully checked throughout.

  1. L35 ROS also includes changed to ROS also include, and things are the same in other places, for example, L73 etc;

Corrected as suggested, and the manuscript carefully checked throughout.

  1. L51water changed to solution; are should be deleted;

Water/solution changed. Can’t see why “are” should be deleted as acting as the verb here.

  1. L89 gene changed to molecular;

Corrected as suggested.

  1. L96 cells there changed to cells, there

Corrected as suggested.

  1. L105-L107 should be rewritten.

This has been reworded and we hope is now more clear.

  1. L148 L261 plants cells should be plant cells

Both lines have been corrected.

  1. L152 heavy metal should be revised, since Al is not heavy metal (Ref51);

Good point, sorry for this slip, and this has therefore been altered.

  1. L176 others too and this too?

The word “too” has been removed in both cases, which reads better.

  1. L205 Superoxide should be Superoxide anion;

Added as suggested.

  1. L231 to should be deleted

Deleted as suggested.

  1. L276-279 L290-L292 L304-L306 should be rewritten

All these sentences have been re-worked for clarity.

Reviewer 2 Report

This review by Hancock and Russell offers a valuable overview on the effects of the signaling from molecular hydrogen. It is indeed well written and highlights key pathways.

I encourage acceptance, not before pursuing these minor amendments:

  • Please formulate concrete questions/goals of the review at the end of the introduction section (line 54). - Authors comment to H2 effects on nitrogen fixation (line 197), but this is just one of many abiotic and biotic factors influencing fixation. In order to avoid being over simplistic, briefly refer to the abiotic-biotic interface by citing the roles of nutrient availability (Oecologia 2016 180(4):1015-24) and accumulation (Front Genet 2020 11:656), soil-plant interactions (Basic Appl Ecol 2014 15(4):305-15), and biotic facilitation (Basic Appl Ecol 2015 16(3):202-9).
  • When referring to the role of molecular hydrogen in abiotic tolerance in plants  (line 230), provide concrete references for drought tolerance (BMC Genet 2012 13:58, Theor Appl Genet 2012 125(5):1069-85, and Plant Sci 2016 242:250-9 for various candidate gene approaches, and PLoS One 2013 8(5):e62898 and Front Plant Sci 2018 9:128  for GxE interactions), heat resistance (J Ecol 2016 104(4):1041-50, and Front Genet 2019 10:954), and responses to frost (Oecologia 2014 175:219-29).
  • When discussing paramagnetic properties and possible cellular effects in section 2.4 in line 242, hypothesize (more as a perspective) on the role of quantum biological processes (mention J R Soc Interface 2018 15:20180640).      

Author Response

Referee 2:

Thank you for the feedback and comments on our manuscript. We are glad you liked it and think that it “offers a valuable oversight”. We have addressed all the points raised, as outlined below. We hope that it is now acceptable for publication.

This review by Hancock and Russell offers a valuable overview on the effects of the signaling from molecular hydrogen. It is indeed well written and highlights key pathways.

I encourage acceptance, not before pursuing these minor amendments:

  • Please formulate concrete questions/goals of the review at the end of the introduction section (line 54).

A couple of sentences have been added and a little re-writing has been carried out to address this.

  • Authors comment to H2 effects on nitrogen fixation (line 197), but this is just one of many abiotic and biotic factors influencing fixation. In order to avoid being over simplistic, briefly refer to the abiotic-biotic interface by citing the roles of nutrient availability (Oecologia 2016 180(4):1015-24) and accumulation (Front Genet 2020 11:656), soil-plant interactions (Basic Appl Ecol 2014 15(4):305-15), and biotic facilitation (Basic Appl Ecol 2015 16(3):202-9).

We have added all these citations as suggested, at the place suggested.

  • When referring to the role of molecular hydrogen in abiotic tolerance in plants  (line 230), provide concrete references for drought tolerance (BMC Genet 2012 13:58, Theor Appl Genet 2012 125(5):1069-85, and Plant Sci 2016 242:250-9 for various candidate gene approaches, and PLoS One 2013 8(5):e62898 and Front Plant Sci 2018 9:128  for GxE interactions), heat resistance (J Ecol 2016 104(4):1041-50, and Front Genet 2019 10:954), and responses to frost (Oecologia 2014 175:219-29).

These references have been added at the place suggested.

  • When discussing paramagnetic properties and possible cellular effects in section 2.4 in line 242, hypothesize (more as a perspective) on the role of quantum biological processes (mention J R Soc Interface 2018 15:20180640).

We have added the paper suggested and also cited a more recent review on the topic by Kim et al. (2021), ref 125.